# Using information theory to optimise epidemic models for real-time prediction and estimation

**Kris V. Parag**[1]*, **Christl A. Donnelly**[1,2]

**1** MRC Centre for Global Infectious Disease Analysis, Imperial College London, London, W2 1PG, United Kingdom, **2** Department of Statistics, University of Oxford, Oxford, OX1 3LB, United Kingdom

* k.parag@imperial.ac.uk

## Abstract

The effective reproduction number, $R_t$, is a key time-varying prognostic for the growth rate of any infectious disease epidemic. Significant changes in $R_t$ can forewarn about new transmissions within a population or predict the efficacy of interventions. Inferring $R_t$ reliably and in real-time from observed time-series of infected (demographic) data is an important problem in population dynamics. The renewal or branching process model is a popular solution that has been applied to Ebola and Zika virus disease outbreaks, among others, and is currently being used to investigate the ongoing COVID-19 pandemic. This model estimates $R_t$ using a heuristically chosen piecewise function. While this facilitates real-time detection of statistically significant $R_t$ changes, inference is highly sensitive to the function choice. Improperly chosen piecewise models might ignore meaningful changes or over-interpret noise-induced ones, yet produce visually reasonable estimates. No principled piecewise selection scheme exists. We develop a practical yet rigorous scheme using the accumulated prediction error (APE) metric from information theory, which deems the model capable of describing the observed data using the fewest bits as most justified. We derive exact posterior prediction distributions for infected population size and integrate these within an APE framework to obtain an exact and reliable method for identifying the piecewise function best supported by available epidemic data. We find that this choice optimises short-term prediction accuracy and can rapidly detect salient fluctuations in $R_t$, and hence the infected population growth rate, in real-time over the course of an unfolding epidemic. Moreover, we emphasise the need for formal selection by exposing how common heuristic choices, which seem sensible, can be misleading. Our APE-based method is easily computed and broadly applicable to statistically similar models found in phylogenetics and macroevolution, for example. Our results explore the relationships among estimate precision, forecast reliability and model complexity.

**Data Availability Statement:** All code and data are available at https://github.com/kpzoo/model-selection-for-epidemic-renewal-models.

**Funding:** KVP and CAD acknowledge joint Centre funding from the UK Medical Research Council and

Department for International Development under grant reference MR/R015600/1. CAD thanks the UK National Institute for Health Research Health Protection Research Unit (NIHR HPRU) in Modelling Methodology at Imperial College London in partnership with Public Health England (PHE) for funding (grant HPRU-2012–10080). The funders had no role in study design, data collection and analysis, decision to publish, or preparation of the manuscript.

**Competing interests:** The authors have declared that no competing interests exist.

## Author summary

Understanding how the population of infected individuals (which may be humans, animals or plants) fluctuates in size over the course of an epidemic is an important problem in epidemiology and ecology. The effective reproduction number, $R$, provides an intuitive and useful way of describing these fluctuations by characterising the growth rate of the infected population. An $R > 1$ signifies a burgeoning epidemic whereas $R < 1$ indicates a declining one. Public health agencies often use $R$ to inform or corroborate vaccination and quarantine policies. However, popular approaches to inferring $R$ from epidemic data make heuristic choices, which may lead to visually reasonable estimates that are deceptive or unreliable. By adapting mathematical tools from information theory, we develop a general and principled scheme for estimating $R$ in a data-justified way. Our method exposes the pitfalls of heuristic estimates and provides an easily computable correction that also maximises our ability to predict upcoming population fluctuations. Our work is widely applicable to similar inference problems found in evolution and genetics, demonstrably useful for reliably analysing emerging epidemics in real time and highlights how abstract mathematical concepts can inspire novel and practical biological solutions, showcasing the importance of multidisciplinary research.

## Introduction

The time-series of newly infected cases (infecteds) observed over the course of an infectious disease epidemic is known as an incidence curve or epi-curve. These curves offer prospective insight into the spread of a disease within an animal or human population by informing on the effective reproduction number, which defines the average number of secondary infections induced by a primary one [1]. This reproduction number, denoted $R_t$ at time $t$, is an important prognostic of the demographic behaviour of an epidemic. If $R_t > 1$, for example, we can expect and hence prepare for exponentially increasing incidence, whereas if $R_t < 1$, we can be reasonably confident that the epidemic has been arrested [1].

Reliably estimating meaningful changes in $R_t$ is an important problem in epidemiology and population biology, since it can forewarn about the growth rate of an outbreak and signify the level of control effort that must be initiated or sustained [2, 3]. While we explicitly consider epidemic applications here, similar reproduction numbers (and growth rates) can be defined for many ecological problems [4], for example in species conservation where we might aim to infer species population dynamics from time-series of sample counts.

The renewal model [5] is a popular approach for inferring salient fluctuations in $R_t$ that is based on the fundamental Euler-Lotka reproduction equation from ecology and evolution [4, 6]. This model has been used to predict Ebola virus disease case counts and assess the transmission potential of pandemic influenza and Zika virus, among others [2, 7, 8]. Currently, it is in widespread use as a tool for tracking the progress of interventions across countries in response to the COVID-19 pandemic [9]. The renewal model may be applied retrospectively to understand the past behaviour of an epidemic or prospectively to gain real-time insight into ongoing outbreak dynamics [5, 10]. We only consider the latter here (see [11] for an investigation of the former).

The prospective approach approximates $R_t$ with a piecewise-constant function i.e. $R_t$ is constant (stable) over some sliding window of $k$ time units (e.g. days or weeks) into the past, beyond which a discontinuous change is assumed [10]. This formulation models the non-stationary nature of epidemics, capturing the idea that different population dynamics are

expected during distinct phases (e.g. onset, growth, control) of the epidemic lifetime. The window length, $k$, is essentially a hypothesis about the stability of $R_t$ underpinning the observed epi-curve. It is critical to reliably characterising the epidemic because it controls the time-scale over which $R_t$ fluctuations are deemed significant.

Too large or small a $k$-value can respectively lead to over-smoothing (which ignores important changes) or to random noise being misinterpreted as meaningful. Inferring under a wrong $k$ can appreciably affect our understanding of an epidemic, as observed in [5], where case reproduction numbers (a common $k$-choice) were found to over-smooth significant changes in HIV transmission, for example. Surprisingly, no principled method for optimising $k$ exists. Current best practice either relies on heuristic choices [7] or provides implicit bounds on $k$ [10]. Here we adapt the minimum description length (MDL) principle from information theory to develop a simple but rigorous selection framework.

The description length is defined as the number of bits needed to communicate a model ($\mathcal{M}$) and the data given that model ($\mathcal{D} \,|\, \mathcal{M}$) on some channel. More complex models increase $L(\mathcal{M})$ (more bits are needed to represent its extra parameters) but decrease $L(\mathcal{D} \,|\, \mathcal{M})$ (as it can better fit the data) for example. Here $L(.)$ indicates length. The MDL model choice minimises $L(\mathcal{M}) + L(\mathcal{D} \,|\, \mathcal{M})$ and is considered the model most justified by the data $\mathcal{D}$ [12]. We adapt an approximation to MDL known as the accumulated prediction error (APE) [13] to identify the $k$ best justified by the available epi-curve, $k^*$ (see the S1 Text for further details and for the general definition of APE).

We analytically derive the posterior predictive incidence distribution of the renewal model, which allows us to evaluate cumulative log-loss prediction scores at any $k$ exactly. The APE choice, $k^*$, minimises these scores and, additionally, optimises short-term prediction accuracy [14]. Our method is valid at all sample sizes, easily computed for arbitrary model dimensions and, unlike many selection criteria of similar computability, includes parametric complexity [12]. Parametric complexity measures how functional relationships among parameters influence complexity, and when ignored (as in the Bayesian or Akaike information criteria) can lead to biased renewal model selection [11].

The APE metric designates the model that best predicts unseen data from the same underlying process, and not the one that best fits existing data, as optimal and of justified complexity [13]. This is equivalent to minimising what is called the generalisation or out-of-sample error in machine learning, and is known to balance under and overfitting. Our APE-based approach therefore pinpoints and characterises only those $R_t$ fluctuations that are integral to achieving reliable short-term incidence growth predictions.

The performance, speed and computational ease of our approach makes it suitable for real-time forecasting and model selection. It could therefore serve as a stand-alone computational tool or be integrated within existing real-time frameworks, such as in [3] or [10], to provide emerging insights into infected population dynamics, or to assess the prospective efficacy of implemented interventions (e.g. vaccination or quarantine). Public health policy decisions or preparedness plans based on improperly specified $k$-windows could be misinformed or over-confident. Our method hopefully limits these risks.

## Methods

### Renewal model window-sizing problem

Let the incidence or number of newly infected cases in an epidemic at present time $t$, be $I_t$. The incidence curve is a historical record of these case counts from the start of the outbreak and is summarised as $I_1^t = \{I_s : 1 \leq s \leq t\}$, with $s$ as a time-indexing variable. For convenience, we assume that incidence is available on a daily scale so that $I_1^t$ is a vector of $t$ daily counts (weeks

or months could be used instead). The associated effective reproduction number and total infectiousness of the epidemic are denoted $R_t$ and $\Lambda_t$, respectively.

Here $R_t$ is the number of secondary cases that are induced, on average, by a single primary case at $t$ [1], while $\Lambda_t$ measures the cumulative impact of past cases, from the epidemic origin at $s = 1$, on the present. The generation time distribution of the epidemic, which describes the elapsed time between a primary and secondary case, controls $\Lambda_t$ (see the S1 Text). In the problems we consider, $I_1^t$ and $\Lambda_1^t$ form our data, the generation time distribution is assumed known, and $R_1^t$ or groupings of this vector are the parameters to be inferred.

The renewal model [5] derives from the classic Euler-Lotka equation [6] (see the S1 Text), and defines the Poisson distributed (Poiss) relationship $I_t \sim \mathrm{Poiss}(R_t\Lambda_t)$. This means that $\mathbb{P}(x) = \frac{1}{x!} e^{-R_t\Lambda_t}(R_t\Lambda_t)^x$, with $x$ indexing possible $I_t$ values. This formulation assumes maximum epidemic non-stationarity (i.e. that demographic transmission statistics change at every time unit) and hence only uses the most recent data ($I_t$, $\Lambda_t$) to infer the current $R_t$. While this maximises the fitting flexibility of the renewal model, it often results in noisy and unreliable estimates that possess many spurious reproduction number changes [7]. Consequently, grouping is employed.

This hypothesises that the reproduction number is constant over a $k$-day window into the past and results in a piecewise-constant function that separates salient fluctuations (changepoints) from negligible ones (the constant segments) [10, 11]. We use $R_{\tau(t)}$ to indicate that the present reproduction number to be inferred is stationary (constant) over the last $k$ points of the incidence curve i.e. the time window $\tau(t) := \{t, t - 1, \ldots, t - k + 1\}$. The data used to estimate $R_{\tau(t)}$ is then ($I_{t-k+1}^t$, $\Lambda_{t-k+1}^t$). This construction allows us to filter noise and increase estimate reliability but elevates bias. In the S1 Text we show precisely how grouping achieves this bias-variance trade-off.

Choosing $k$ therefore amounts to selecting a belief about the scale over which the epidemic statistics are meaningfully varying and can significantly influence our understanding of the population dynamics of the outbreak. Thus, it is necessary to find a principled method for balancing $k$. Ultimately, we want to find a $k$, denoted $k^*$, that (i) is best supported by the epi-curve and (ii) maximises our confidence in making real-time, short-term predictions that can inform prospective epidemic responses. This is no trivial task as $k^*$ would be sensitive to both the specifically observed stochastic incidence of an epidemic and to how past infections propagate forward in time.

We solve (i)-(ii) by applying the accumulated prediction error (APE) metric from information theory [13], which values models on their capacity to predict unseen data from the generating process instead of their ability to fit existing data [14]. The properties and mathematical definition of the APE are provided in the S1 Text and in the subsequent section. The APE uses the window of data preceding time $s$, ($I_{s-k+1}^s$, $\Lambda_{s-k+1}^s$), to predict the incidence at $s + 1$ and assigns a log-score to this prediction. This procedure is repeated over $s \le t$ and for possible $k$-values. The $k$ achieving the minimum cumulated log-score is deemed optimal. Fig 1 summarises and illustrates the APE algorithm.

## Exact model selection using predictive distributions

To adapt APE, we require the posterior predictive incidence distribution of the renewal model, which at time $s$ is $\mathbb{P}(x \mid I_{s-k+1}^s)$, with $x$ indexing the space of possible one-step-ahead predictions at $s + 1$, and $\hat{I}_{s+1} = \mathbb{E}[x]$. We assume a gamma conjugate prior distribution on $R_{\tau(s)}$ as is commonly done in the renewal model frameworks of [3, 10] and because the gamma distribution can fit many unimodal variables on the positive real line. For some hyperparameters $a$ and $c$ this is $R_{\tau(s)} \sim \mathrm{Gam}(a, \frac{1}{c})$, with Gam as a shape-scale parametrised gamma distribution.

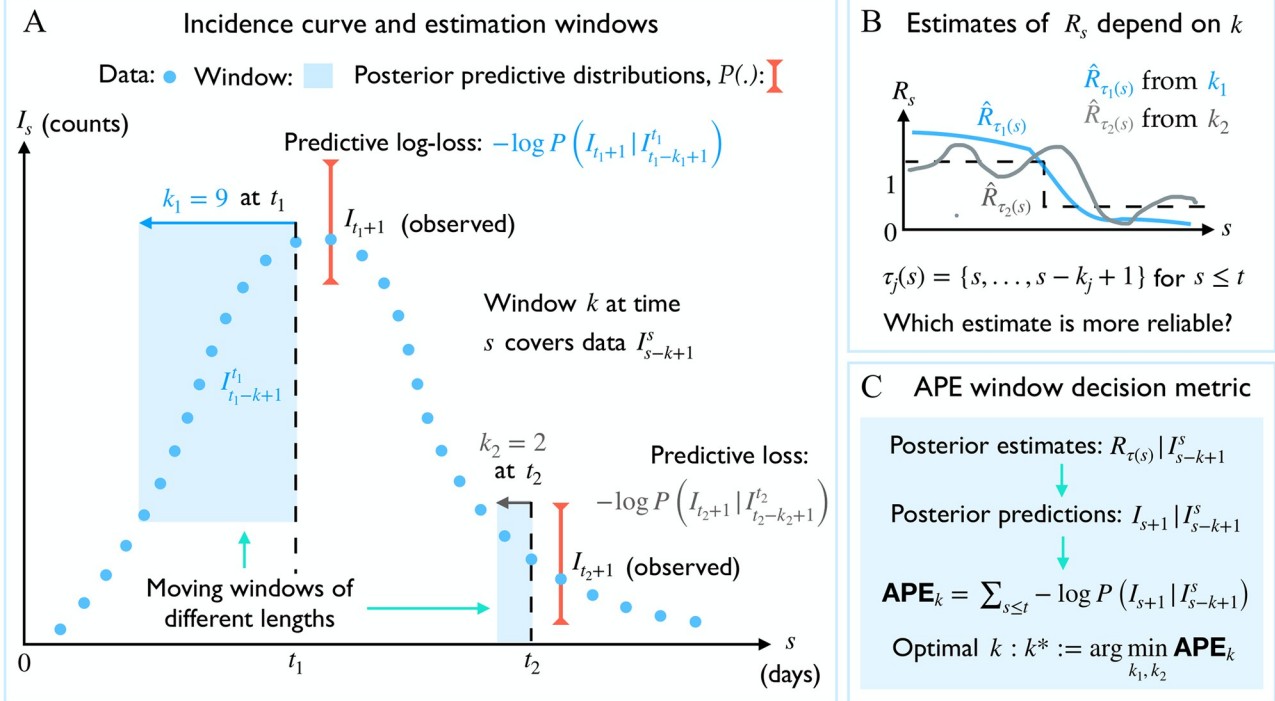

**Fig 1. Optimal window selection using APE.** (A) An observed incidence curve (blue dots) is sequentially and causally predicted over time $s \leq t$ using effective reproduction number estimates based on two possible windows lengths of $k_1$ and $k_2$ (blue shaded). Predictive distributions are summarised by red error bars (shown only for times $t_1$ and $t_2$, respectively for $k_1$ and $k_2$), (B) The true reproduction number ($R_s$, dashed black) is estimated under each window length as $\hat{R}_{\tau_1(s)}$ (blue) and $\hat{R}_{\tau_2(s)}$ (grey). Large windows ($k_1$) smooth over fluctuations. Small ones ($k_2$) recover more changes but are noisy. (C) The APE assesses $k_1$ and $k_2$ via the log-loss of their sequential predictions (i.e. from red error bars across time). The window with the smaller APE is better supported by this incidence curve. See Methods for more mathematical details.

The posterior distribution of $R_{\tau(s)}$, $\mathbb{P}(R_{\tau(s)} \mid I_{s-k+1}^s)$, is then

$$R_{\tau(s)} \mid I_{s-k+1}^s \sim \text{Gam}\left(a + i_{\tau(s)}, \frac{1}{c + \lambda_{\tau(s)}}\right). \tag{1}$$

For convenience we define $\alpha_{\tau(s)} := a + i_{\tau(s)}$, $\beta_{\tau(s)} := {}^1/_{c+\lambda_{\tau(s)}}$ with $i_{\tau(s)}$ and $\lambda_{\tau(s)}$ are the sum of incidence ($I_s$) and total infectiousness ($\Lambda_s$) over the window $\tau(s)$ (see the S1 Text). If a variable $y \sim \text{Gam}(\alpha, \beta)$ then $\mathbb{P}(y) = {}^{y^{\alpha-1}e^{-y/\beta}}/_{\beta^\alpha\Gamma(\alpha)}$ and $\mathbb{E}[y] = \alpha\beta$. The posterior mean estimate is therefore $\hat{R}_{\tau(t)} = \alpha_{\tau(t)}\beta_{\tau(t)}$. Applying Bayes formula and marginalising yields the posterior predictive distribution of the number of infecteds at $s + 1$ as

$$\mathbb{P}(x \mid I_{s-k+1}^s) = \int_0^\infty \mathbb{P}(x \mid R_{\tau(s)})\mathbb{P}(R_{\tau(s)} \mid I_{s-k+1}^s)\, dR_{\tau(s)}. \tag{2}$$

In Eq (2) we used the conditional independence of future incidence data from the past epi-curve, given the reproduction number to reduce $\mathbb{P}(x \mid I_{s-k+1}^s, R_{\tau(s)})$ to $\mathbb{P}(x \mid R_{\tau(s)})$, which expresses the renewal model relation $x \sim \text{Poiss}(R_{\tau(s)}\Lambda_{s+1})$. As $\Lambda_{s+1}$ only depends on $I_{s-k+1}^s$ there

are no unknowns. Solving using this and Eq (1) gives $\mathbb{P}(x \mid I_{s-k+1}^s) = \phi_1 \phi_2^{-1}$ where

$$\phi_1 := \Lambda_{s+1}^x \int_0^\infty R_{\tau(s)}^{x+\alpha_{\tau(s)}-1} e^{-R_{\tau(s)}\left(\Lambda_{s+1}+\frac{1}{\beta_{\tau(s)}}\right)} \, dR_{\tau(s)} \text{ and}$$

$$\phi_2 := \beta_{\tau(s)}^{\alpha_{\tau(s)}} \Gamma(x+1)\Gamma(\alpha_{\tau(s)}), \text{ with } \Gamma(z) := \int_0^\infty \theta^{z-1}e^{-z} \, d\theta.$$

(3)

Since $\int \theta^{z-1}e^{-y\theta} \, d\theta \equiv {}^{\Gamma(z)}/_{y^z}$ the integral in $\phi_1$ simplifies to $\frac{\Gamma(x+\alpha_{\tau(s)})}{(\Lambda_{s+1}+{}^1/_{\beta_{\tau(s)}})^{x+\alpha_{\tau(s)}}}$. Some algebra then reveals the negative binomial (NB) distribution:

$$x \mid I_{s-k+1}^s \sim \text{NB}\left(\alpha_{\tau(s)}, p_{\tau(s)} := \frac{\Lambda_{s+1}\beta_{\tau(s)}}{1+\Lambda_{s+1}\beta_{\tau(s)}}\right).$$

(4)

If some variable $y \sim \text{NB}(\alpha, p)$ then $\mathbb{P}(y) := \binom{\alpha+y-1}{y}(1-p)^\alpha p^y$ and $\mathbb{E}[y] = {}^{p\alpha}/_{1-p}$. Eq (4) is a key result and relates to a framework developed in [3]. It completely describes the one-step-ahead prediction uncertainty and has mean $\mathbb{E}[x] = \hat{I}_{s+1} = \Lambda_{s+1}\hat{R}_{\tau(s)}$, and variance $\text{var}(x) = \hat{I}_{s+1}\left(1 + \frac{\hat{I}_{s+1}}{i_{\tau(s)}+a}\right)$. These relations explicate how the current estimate of $R_\tau(s)$ influences our ability to predict upcoming incidence points. At $s = t$ the above expressions yield prediction statistics for the next (unobserved) time-point beyond the present.

The APE metric is generally defined as $\text{APE}_k := \sum_{s=1}^{t-1} -\log \mathbb{P}(I_{s+1} \mid I_{s-k+1}^s)$ [13] (see the S1 Text) and is controlled by the shape of the posterior predictive distribution over time. We specialise this metric for renewal models and integrate Eq (4) into the algorithm of Fig 1 to derive

$$\text{APE}_k = \sum_{s=1}^{t-1} B_{s+1} + I_{s+1} \log p_{\tau(s)} + \alpha_{\tau(s)} \log(1 - p_{\tau(s)}),$$

(5)

with $I_{s+1}$ as the (true) observed incidence at time $s + 1$, which is evaluated within the context of the predictive space of $x$, and $B_{s+1} := \log\binom{I_{s+1}+\alpha_{\tau(s)}-1}{I_{s+1}}$. Eq (5) is exact, easy to evaluate and offers a simple and direct means of finding the optimal data-justified window length as $k^* := \arg\min_k \text{APE}_k$.

The summation in Eq (5) also clarifies why our approach is useful for real-time applications. As an epidemic unfolds and more data accumulate the upper limit on $s$ will increase. We can update $\text{APE}_k$ to account for emerging data by simply including an additional term for each new data point on top of the previously calculated sum. For example, a new datum at $t + 1$ requires the addition of $B_{t+1} + I_{t+1} \log p_{\tau(t)} + \alpha_{\tau(t)} \log(1 - p_{\tau(t)})$ to Eq (5). In the Results section we will demonstrate this by successively computing $k_s^*$, the optimal window length for data up to time $s$.

$\text{APE}_k$ can also be computed using the in-built NB routines of many software (some parametrise this distribution differently so that Eq (4) may need to be implemented as $\text{NB}(\alpha_{\tau(s)}, 1 - p_{\tau(s)})$). When computing Eq (5) directly, the most difficult term is $B_{s+1}$ when $I_{s+1}$ and $\alpha_{\tau(s)}$ are large. In these cases Stirling approximations may be applied. We provide Matlab and R implementations of our renewal model APE metric at https://github.com/kpzoo/model-selection-for-epidemic-renewal-models.

## Results

### Optimal window selection for dynamic outbreaks

We apply our APE metric to select $k^*$ for several epidemic examples, which examine reproduction number profiles featuring stable (Fig 2A) and seasonally fluctuating (Fig 2B) transmission, as well as exponential (Fig 3A) and step-changing (Fig 3B)) control measures. These examples explore dynamically diverse epidemic scenarios and consider both large (Figs 2A and 3A) and small outbreaks (Figs 2B and 3B). Small outbreaks are fundamentally more difficult to estimate. We simulated epidemics for $t = 200$ days using a generation time distribution analogous to that used for Ebola virus disease predictions in [15]. We investigated a window search space of $2 \leq k \leq \frac{t}{2}$ and computed the APE at each $k$, over the epidemic duration ($1 \leq s \leq t$).

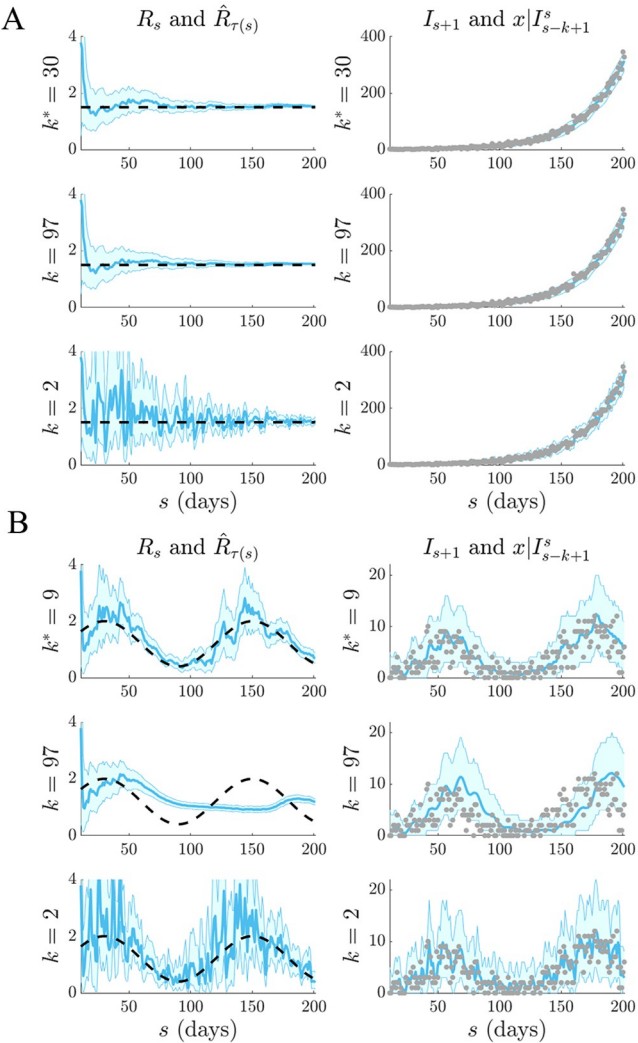

**Fig 2. Selection for stable and fluctuating epidemics.** Left graphs compare $\hat{R}_{\tau(s)}$ estimates (blue with 95% confidence intervals) at the APE window length $k^*$ to those when $k$ is set to its upper and lower limits. Right graphs give corresponding one-step-ahead predictions $\hat{I}_s$ given the window of data $I_{\tau(s-1)}$ (blue with 95% prediction intervals). Dashed lines are the true $R_s$ numbers (left) and dots are the true $I_s$ counts (right). The panels examine (A) stable (constant) and (B) periodically varying changes in $R_s$.

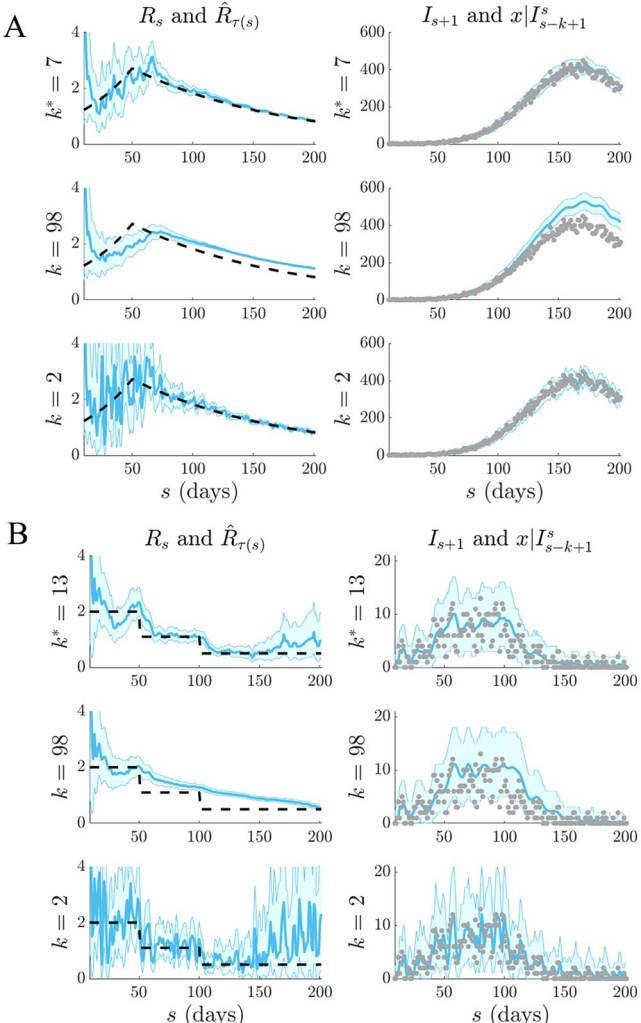

**Fig 3. Selection for epidemics with interventions.** Left graphs compare $\hat{R}_{\tau(s)}$ estimates (blue with 95% confidence intervals) at the APE window length $k^*$ to those when $k$ is set to its upper or lower limits. Right graphs give corresponding one-step-ahead predictions $\hat{I}_s$ given the window of data $I_{\tau(s-1)}$ (blue with 95% prediction intervals). Dashed lines are the true $R_s$ numbers (left) and dots are the true $I_s$ counts (right). The panels examine (A) exponentially rising and decaying and (B) piecewise falling $R_s$ due to differing interventions.

Reproduction number estimates, $\hat{R}_{\tau(s)}$, and demographic incidence predictions $x \mid I_{s-k+1}^s$, are presented in Figs 2 and 3. Predictions over the first $s < k$ times use all $s$ data points.

We find that the APE metric balances $\hat{R}_{\tau(s)}$ estimate accuracy against one-step-ahead predictive coverage (i.e. the proportion of time that the true $I_{s+1}$ lies within the 95% prediction intervals of $x \mid I_{s-k+1}^s$) for a range of $R_s$ dynamics (top graphs of Figs 2 and 3). This dual optimisation is central to this work. Moreover, vastly different $I_{s+1}$ predictions and $\hat{R}_{\tau(s)}$ estimates can result when $k$ is misspecified (middle and bottom graphs). This can be especially misleading when attempting to identify the significant changes in epidemic transmissibility and can support substantially different beliefs about the infection population biology. Estimation and prediction performance also depend on the actual incidence at any time; small $I_s$ values lead to wider confidence intervals in $\hat{R}_{\tau(s)}$ at any $k$ (see Eq. (S3) of the S1 Text).

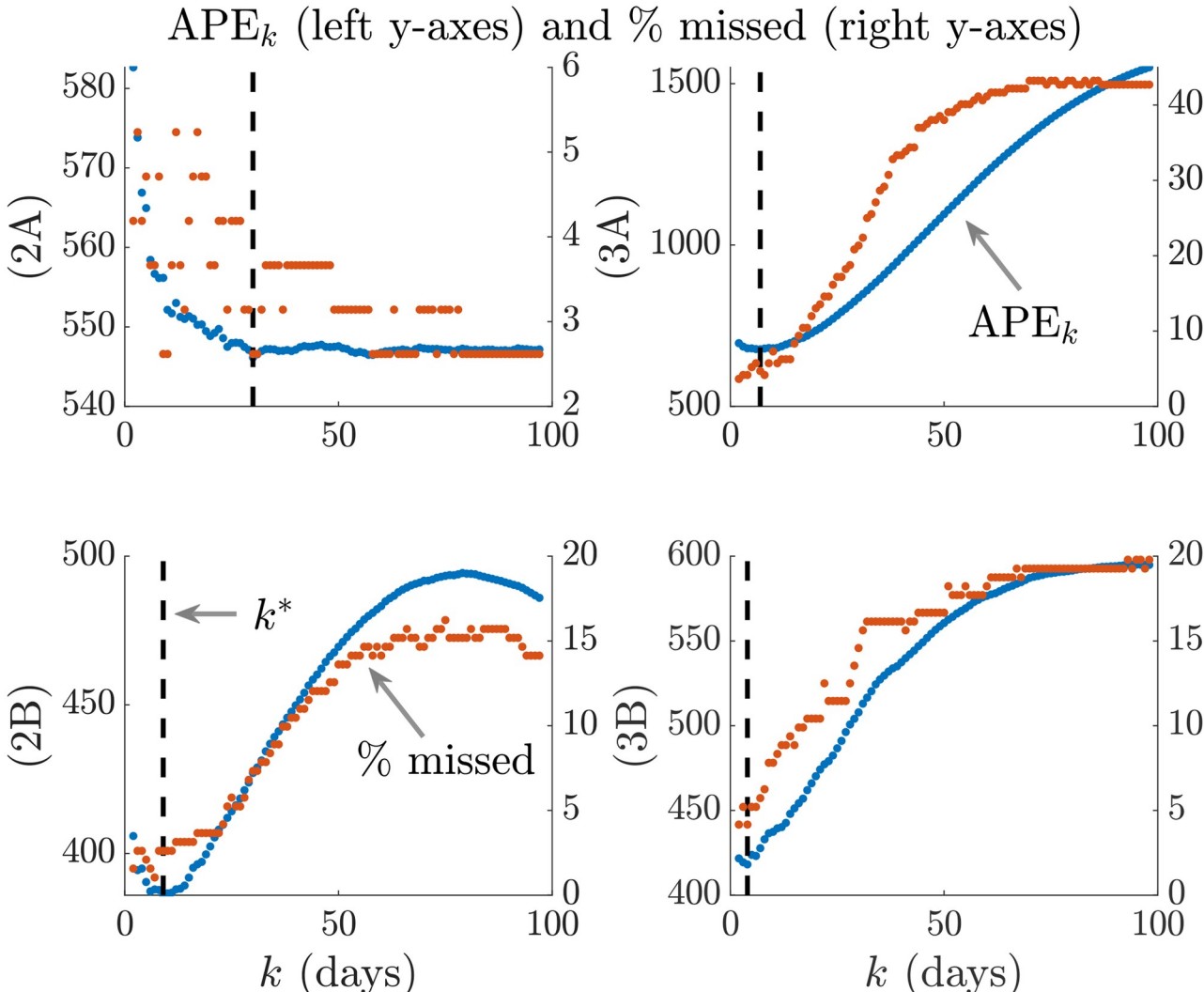

**Fig 4. APE prediction accuracy.** We compare the APE metric (blue, left y axes) to the percentage of true incidence values, $I_{s+1}$ that fall outside the 95% prediction intervals of $x \mid I_{s-k+1}^s$ (red, right y axes) for various window sizes $k$. The dashed line is $k^*$. Panels correspond to those of Figs 2 and 3.

When $k$ is unjustifiably large not only do we observe systematic prediction and estimation errors, but alarmingly, we tend to be overconfident in them. When $k$ is too small, we infer rapidly and randomly fluctuating $\hat{R}_{\tau(s)}$ values, which sometimes can deceptively underlie reasonably looking incidence predictions. Consequently, optimal $k$-selection is integral for trustworthy inference and prediction. Observe that small $k$, which implies a more complex renewal model (i.e. there are more parameters to be inferred), does not generally result in better causally predictive one-step-ahead incidence predictions. Had we instead naively picked the $k$ that best fits the existing epi-curve, then the smallest $k$ would always be favoured (overfitting) [11].

We emphasize and validate the predictive performance of APE in Fig 4. This shows, for all simulated examples above that minimising the APE also approximately minimises the percentage of the time $s \leq t$ that true incidence values fall outside the 95% prediction intervals of Eq (4). This validates the APE approach and evidences its proficiency at optimising for short-term forecasting accuracy in real time. We recommend using APE to define $k^*$ for an existing epi-

curve up to the present $t$. Applying Eq (4) with this $k^*$ should then best predict the number of cases on the $(t + 1)^{th}$ day. This entire procedure should be repeated for later forecasts, with $k^*$ being progressively updated. The real-time behaviour of $k^*$ as data accumulate over the course of an outbreak is investigated in the next section.

## Real-time performance for emerging outbreaks

During an unfolding epidemic, renewal models can support response efforts by providing real-time estimates of infectious disease transmissibility and short-term forecasts of the expected incidence of cases [9]. Here we show how the APE metric can be used within this framework as a rapid and reliable real-time tool. We focus on the key problem of diagnosing and verifying the efficacy of implemented control measures over the course of an emerging outbreak [3]. Knowing whether an intervention is working or not can inform preparedness and resource allocation. Estimating whether $R_s < 1$ or $R_s \geq 1$ is one simple means of assessing efficacy. Effective control measures lead to the former.

We investigate how the APE metric responds in real time to swift swings in reproduction number and quantify our results over $10^3$ simulated epi-curves with $t = 150$ days. We focus on step-changes in $R_s$ and examine how the successively optimal window length, denoted $k_s^*$ when computed with data up to time $s$, responds. If $k_s^*$ is sensitive to these changes with minimum delay then it is likely a dependable means of diagnosing control efficacy in real time. In previous sections we have been computing $k^* = k_t^*$. We consider four models and present our main results in Figs 5 and 6.

In Fig 5 we examine cases of increasing transmission for (A) an outbreak that is uncontrolled and worsening (e.g. if the infectious disease acquires an additional route of spread) and (B) an ineffectively controlled epidemic with a late-stage elevation in $R_s$ (e.g. if quarantine measures were relaxed too quickly). Fig 6 then investigates scenarios involving effective interventions where (A) rapid control is employed (e.g. if a wide-scale lockdown is enacted) and (B) the initial action is only partially effective and so later additional controls are needed (e.g. if flight restrictions were first applied but then further transport shutdowns or mobility restrictions were required).

In both figures, top graphs overlay true $I_{s+1}$ and predicted $x \mid I_{s+k-1}^s$ across time and middle ones show the best $\hat{R}_{\tau(s)}$ estimates under the final $k_t^*$. We see that the APE-selected model properly distinguishes significant changes from stable periods in all scenarios. The model in Fig 5A is the most difficult to infer as the increasing $R_s$ does not visibly change the shape of the incidence population curve. Bottom graphs show $k_s^*$ sequentially across the ongoing epidemic. Here we observe how the APE metric uses data to justify its choices in real time. As data accumulate under a stable reproduction number APE increases $k^*$. This makes sense as there is increasing support for stationary epidemic behaviour. This is especially obvious in Fig 6A.

However, on facing a step-change the APE immediately responds by drastically reducing $k_s^*$ – both in scenarios where $R_s$ is changing from above to below 1 and vice versa. This reduction is less visible in Fig 5A because the observed epi-curve is not as dramatically altered as in the other scenarios. This rapid response recommends the APE as a suitable and sensitive real-time diagnostic tool. We dissect the impact of change-times on the actual APE values at different $k$ in Fig 7. There we find that when the epi-curve is reasonably stable there is not much difference between the performance at various window lengths and so the APE curves are neighbouring. In contrast, when a non-stationary change occurs there is a clear unravelling of the curves and potentially large gains to be made by optimising $k$. Failing to properly select $k$ here would potentially mislead our understanding of the unfolding outbreak. This increases the impetus for formal $k$-selection metrics such as the APE.

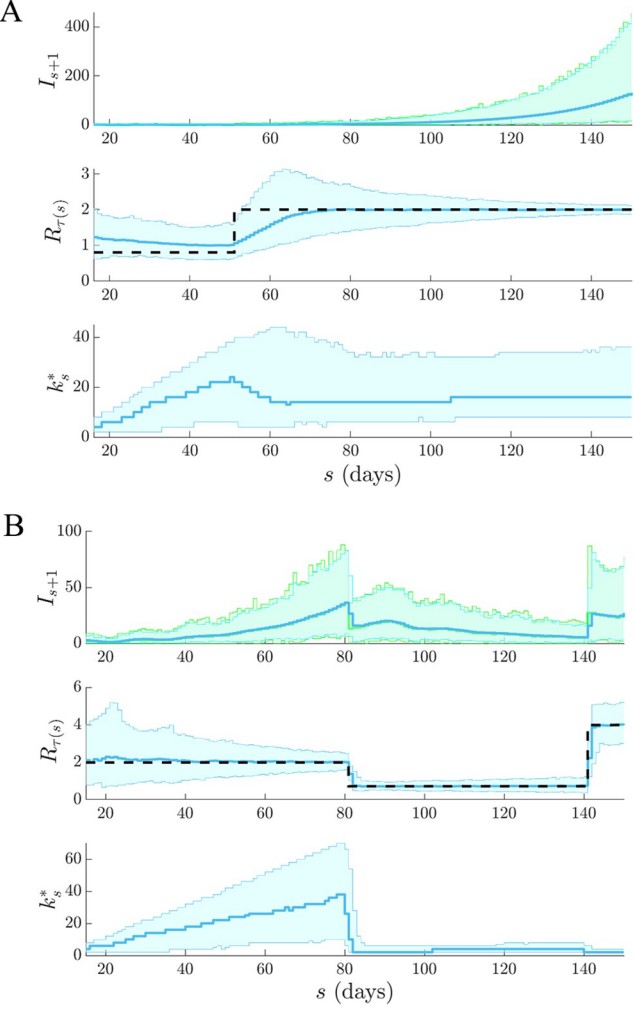

**Fig 5. Real-time APE sensitivity to increasing transmission.** We simulate $10^3$ independent epi-curves under renewal models with sharply (A) increasing and (B) recovering epidemics. Top graphs give the true (green) and predicted (blue) incidence ranges, the middle ones provide estimates of $R_s$ under the final $k_t^*$ and the bottom graphs illustrate how successive $k_s^*$ choices from APE vary across time and are sensitive to real-time elevations in transmission.

## Window selection on empirical data

We test our APE approach on two well-studied empirical epidemic datasets: pandemic H1N1 influenza in Baltimore from 1918 [16] and SARS in Hong Kong from 2003 [17]. For each epidemic we extract the incidence curve, total infectiousness and generation time distributions from the EpiEstim R package [10] and apply the APE metric over $2 \leq k \leq {}^t/_2$ with $t$ as the last available incidence time point. We compare the one-step-ahead $I_{s+1}$ prediction fidelity and the $R_{\tau(s)}$ estimation accuracy obtained from the renewal model under the APE-selected $k^*$ to that from the model used in [10], which recommended weekly windows (i.e. $k = 7$) after visually examining several window lengths.

Our main results are in Figs 8–10. We benchmarked our estimates against those directly provided by EpiEstim to confirm our implementation and restrict $k > 1$ to avoid model identifiability issues [11, 18]. Intriguingly, we find $k^* = \min k = 2$ for both datasets. This yields appreciably improved prediction fidelity (i.e. the coverage of observed incidence values by the

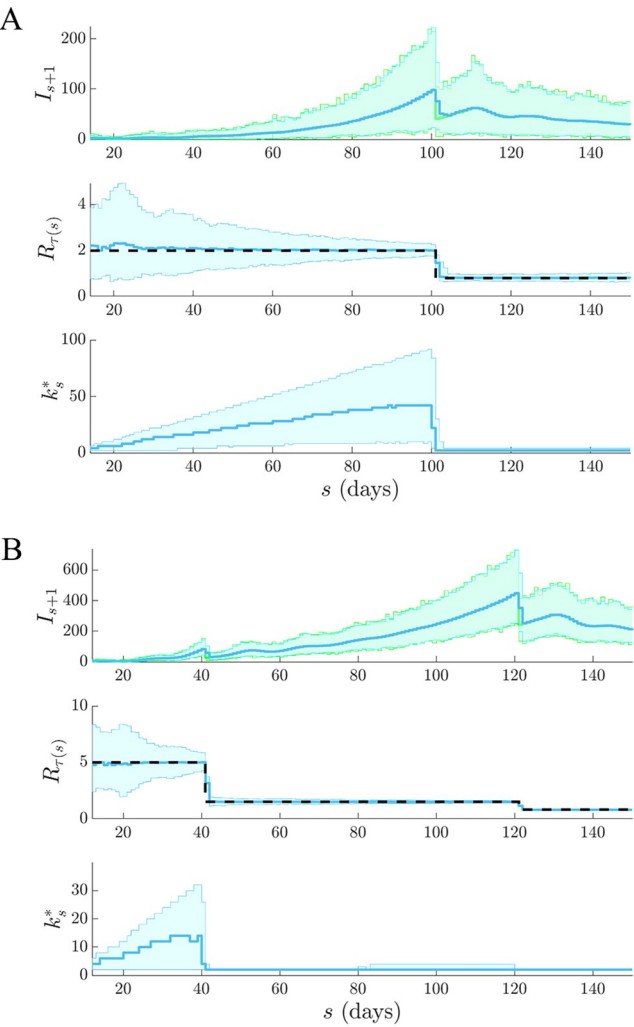

**Fig 6. Real-time APE sensitivity to rapid epidemic control.** We simulate $10^3$ independent epi-curves under renewal models with (A) one effective and (B) two partially effective interventions. Top graphs give the true (green) and predicted (blue) incidence ranges, the middle ones provide estimates of $R_s$ under the final $k_i^*$ and the bottom graphs illustrate how successive $k_s^*$ choices from APE can reliably and rapidly detect the impact of real-time control actions.

95% prediction intervals), relative to the weekly window choice, as seen in Fig 8. Between 8 −10% of incidence data points are better covered by using $k^*$ over $k = 7$. This improvement is apparent in the right graphs of Figs 9A and 10A, where the $k = 7$ case produces stiffer incidence predictions that cannot properly reproduce the observed epi-curve.

Weekly windows misjudge the SARS epidemic peak, predict a multimodal SARS incidence curve that is not reflected by the actual data and systematically underestimate influenza case counts when it matters most (i.e. around the high incidence phase). However, the smaller $k^*$ window results in noisier $R_{\tau(s)}$ estimates (left graphs of Figs 9A and 10A, which may be questionable. These rapidly fluctuating reproduction numbers likely motivated the adoption of weekly windows in previous analyses. While the APE-based estimates are indeed more uncertain (a consequence of shorter windows), we argue that they are formally justified by the available data, especially given the weaker predictive capacity of weekly windows. While the noisier $R_{\tau(s)}$ estimates do not mislead our understanding of the efficacy of implemented control

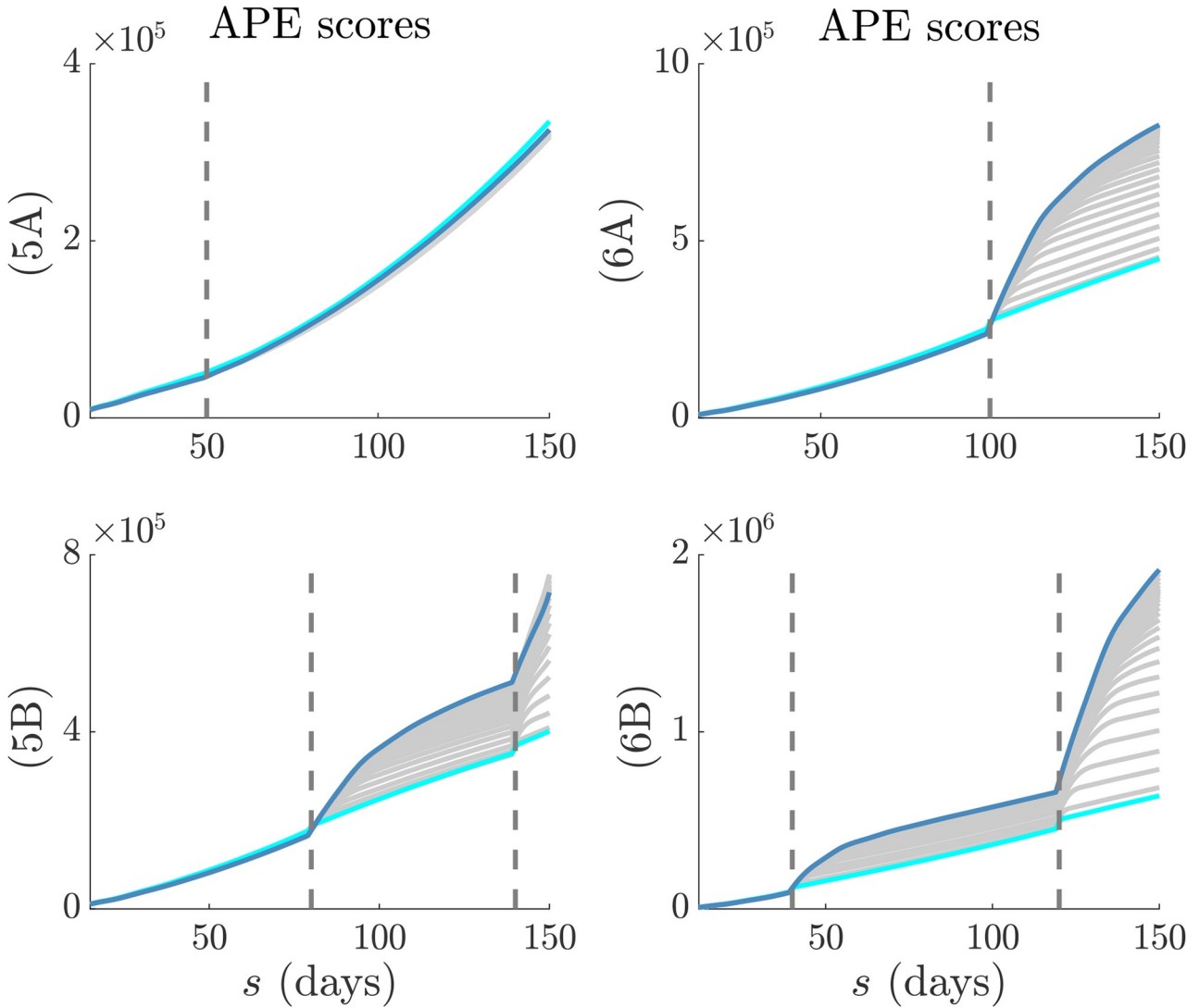

**Fig 7. APE scores in real time.** The successive (in time) APE scores for each window length, *k*, are shown in grey. The scores for the smallest and largest *k* are in cyan and magenta respectively. Graphs correspond to the models in Figs 5 and 6. Dashed lines are the change-times of each model. In (5A) the change-time does not significantly affect the epi-curve shape and so the APE scores are close together. In (5B), (6A) and (6B) the change-times notably alter the epi-curve shape so that the choice of *k* becomes critical to performance.

measures (it is still clear that the influenza epidemic is only partially under control between $40 \leq s \leq 65$ days before recovering, while the SARS outbreak is largely arrested from $s > 50$ days), they likely overestimate the peak transmissibility of these diseases.

We might suspect that certain artefacts of the data could resolve this issue, rendering a more believable combination of estimated reproduction number and predicted incidence. Particularly, the influenza data seem considerably more affected by the smaller look-back windows. These $k^* = 2$ windows are needed to help predictions get close to the peak incidence values of the data. However, these peaks seem reminiscent of outliers and in the original analysis of [16] they were attributed to possible recollection bias in patients that were questioned. Removing these biases might be expected to lead to smoother APE-justified reproduction numbers. We test this hypothesis in Figs 9B and 10B.

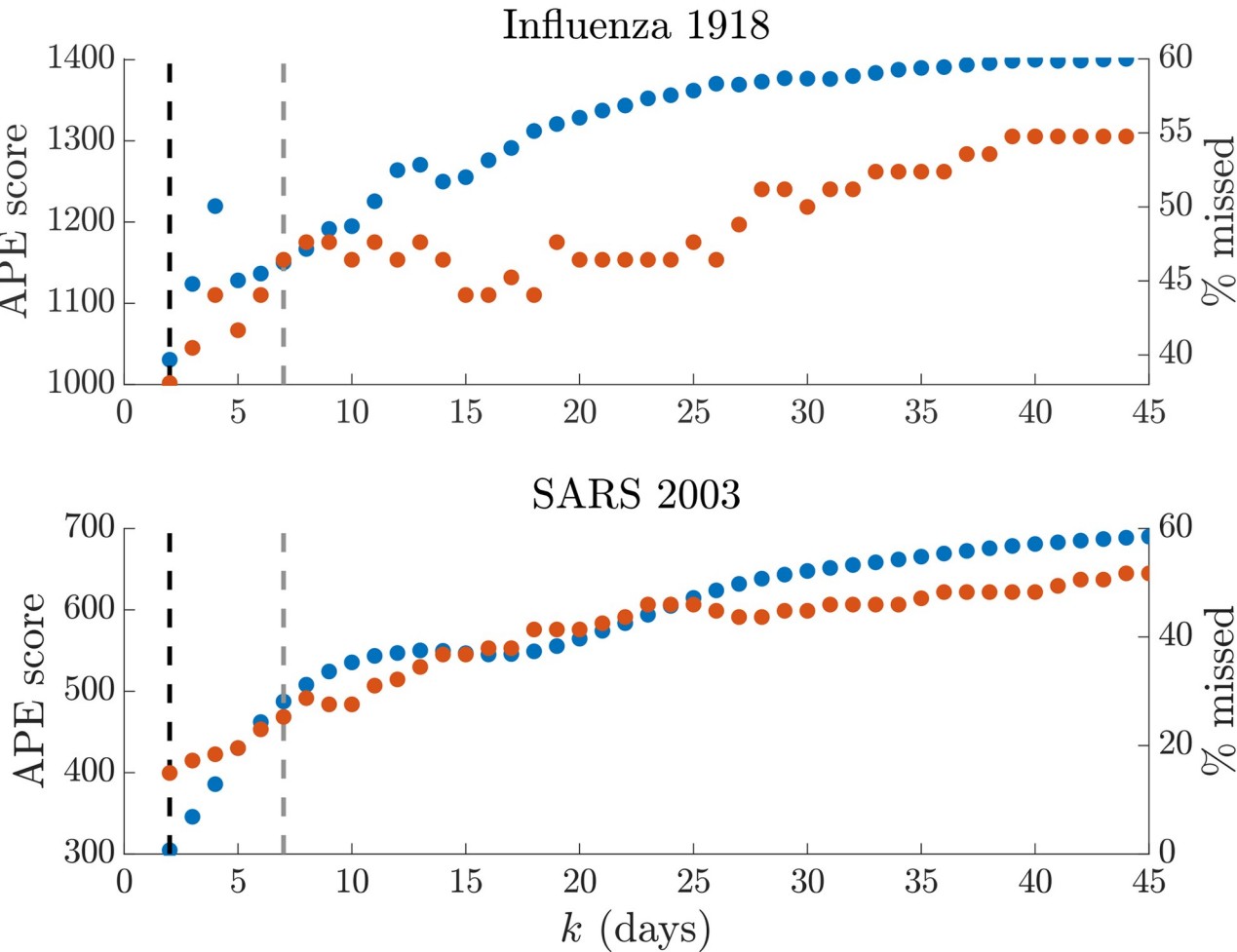

**Fig 8. Empirical prediction accuracy.** We compare the APE metric (dotted blue, left y axis) to the percentage of true incidence values, $I_{s+1}$, which fall outside the 95% prediction intervals of $x \mid I^s_{s-k+1}$ (dotted red, right y axis) across the window search space $k$. The dashed line gives $k^*$ (black) and $k = 7$ (grey). The top graph presents results for the influenza 1918 dataset, while the bottom one is for SARS 2003 data. We find that heuristic weekly windows lead to appreciably larger forecasting error than the APE selections.

There we find that applying simple 5-day moving average filters to the data, as done in [16] to ameliorate outliers, still does not support a $k^* > 2$ in either scenario, though smoother estimates do result. Weekly windows are still too inflexible to properly predict this averaged incidence. While other signal processing techniques, such as using autocorrelated smoothing prior distributions instead of independent gamma ones over the reproduction numbers, could be applied to further investigate if $k^* > 2$ is justifiable we consider this beyond the scope of this work and somewhat biologically unmotivated. Instead we conjecture that these results support the interesting alternative hypothesis that the epi-curve population data are not Poisson distributed. We expand on this conjecture in the subsequent section.

## Discussion

Inferring the dynamics of the effective reproduction number, $R_t$, in real time is crucial for forecasting transmissibility and the efficacy of implemented control actions and for assessing the growth of an unfolding infectious disease outbreak [3]. Renewal models provide a popular

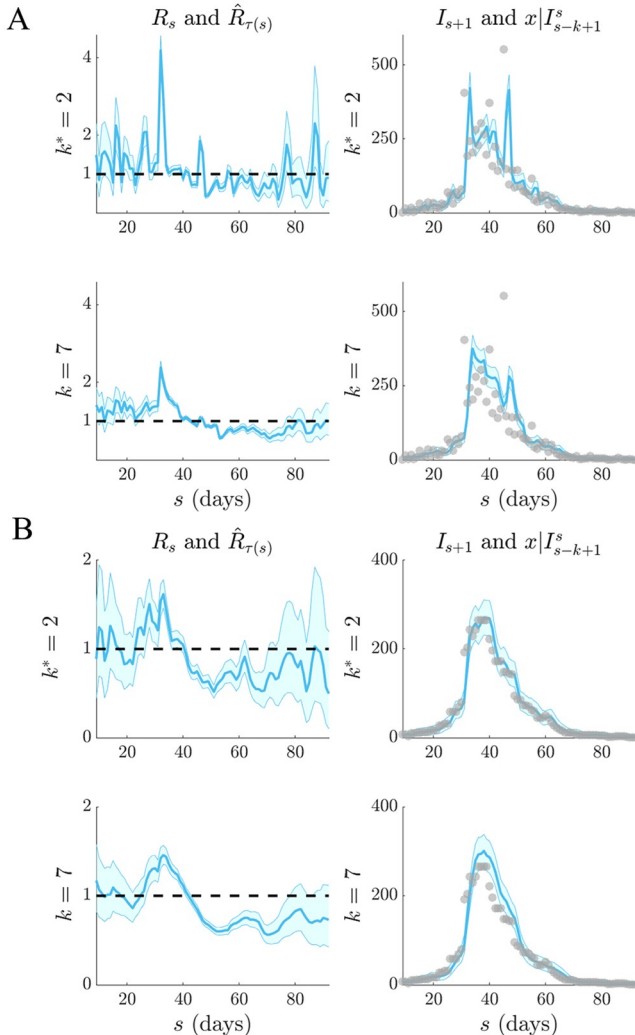

**Fig 9. Selection for pandemic influenza (1918).** Left graphs compare $\hat{R}_{\tau(s)}$ estimates (blue with 95% confidence intervals) at optimal APE window length $k^*$ to weekly sliding windows ($k = 7$), which were recommended in [10]. Right graphs give corresponding one-step-ahead predictions $x \mid I_{s-k+1}^s$ (blue with 95% prediction intervals). Dashed lines are the $R = 1$ threshold (left) and dots are the true incidence $I_s$ (right). Panel (A) directly uses the empirical influenza (1918) data [10] while (B) smooths outliers in the data as in [16].

platform for prospectively estimating these reproduction numbers, which can then be used to generate incidence projections [2, 7, 15]. However, the dependence of these estimates and predictions on the look-back window size, $k$, which determines the renewal model dimensionality, has never been formally investigated.

Previous methods for selecting $k$ have generally been heuristic or ad-hoc [10]. Here we have devised and validated a new, rigorous, information-theoretic approach for optimising $k$ to available incidence data. Our method was founded on deriving an analytical expression for the renewal model posterior predictive distribution. Integrating this into the general APE metric of [13] led to Eq (5), a simple, easily-computed yet theoretically justified metric for $k$-selection that balances estimation fidelity with prediction accuracy.

This APE approach not only accounts for parametric complexity and is formally linked to MDL (and Bayesian model selection − see the S1 Text), but it also has several desirable

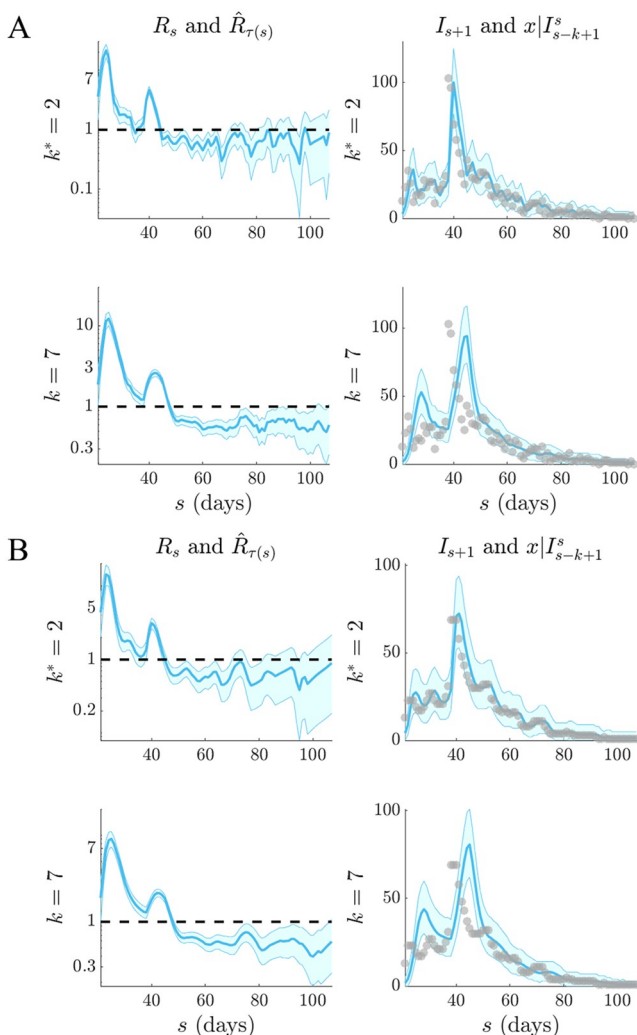

**Fig 10. Selection for SARS (2003).** Left graphs compare $\hat{R}_{\tau(s)}$ estimates (blue with 95% confidence intervals) at optimal APE window length $k^*$ to weekly sliding windows ($k = 7$), which were recommended in [10]. Right graphs give corresponding one-step-ahead predictions $x \mid I^s_{s-k+1}$ (blue with 95% prediction intervals). Dashed lines are the $R = 1$ threshold (left) and dots are the true incidence $I_s$ (right). Panel (A) directly uses the empirical SARS (2003) data [10] while (B) smooths outliers (with a 5-day moving average) in the data as in [16].

properties that render it suitable for handling the eccentricities of real-time infectious disease applications:

(a) Small outbreak sample size. Early-on in an epidemic data are scarce and uncertainty is large. The APE approach is valid and computable at all sample sizes at which the renewal model is identifiable and hence is applicable to emerging outbreaks [14]. Its emphasis on using the full predictive distribution and its Bayesian formulation allow it to properly account for large uncertainties and to handle different (prior) hypotheses about $R_t$, as the epidemic progresses from small (e.g. onset) to large (e.g. establishment) data regimes.

(b) Non-stationary transmission. Incidence time series can change rapidly across an unfolding outbreak and hence are not independent and identically distributed [7]. Instead they are sequential, autocorrelated and possess non-stationary (time-varying) statistics. The APE formally accounts for these properties. Leave-one-out cross validation (CV) is a popular model selection

approach that is related to APE. It can be defined as $\text{CV} := \sum_{s=1}^{t-1} -\log \mathbb{P}(I_{s+1} \mid I_{s+1}^{\text{c}})$, where $I_{s+1}^{\text{c}}$ means that all incidence points except $I_{s+1}$ are included. Comparing this to Eq. (S4) of the S1 Text we see that APE is a modified CV that is explicitly specialised for non-stationary, accumulating time-series.

(c) Lack of a 'true model'. It is highly unlikely that the underlying $R_t$ is truly described by one of the piecewise-constant functions assumed within renewal models. Unlike many model selection criteria, the APE does not require a true model to exist within the set being evaluated [12]. It is only interested in finding the model that best predicts the data and so emphasises accurate forecasting and data-justified complexity. However, should a true model exist, the APE is statistically consistent i.e. as data accumulate it will select the data-generating model with probability 1 [12].

While we have specialised our results to epidemiology, our method is widely-applicable. Several popular models in macroevolution, phylogeography and genetics, such as the skyline plot, structured coalescent and sequential Markovian coalescent, all possess piecewise-Poisson statistical formulations analogous to the renewal model [18]. As shown in [11], this leads to an almost plug-and-play usability of MDL-based methodology. Moreover, the renewal model itself can be adapted to other animal and plant ecology problems where the interest is to infer a demographic growth rate from a time-series of species samples [4].

We tested our method on both simulated and empirical datasets. We started by examining various distinct, time-varying reproduction number profiles in Figs 2 and 3, for which no true model existed. By comparing the APE-optimised $k^*$ against long and short window sizes, we found that not only does APE meaningfully balance $R_t$ estimates to achieve good prediction capacity, but also that getting $k$ wrong could promote strikingly different conclusions about the infection population dynamics from the same dataset. This behaviour held consistent for outbreaks with both small and large infected case numbers, stable and seasonal transmission and under different dynamics of control.

We then investigated several step-changing reproduction number examples in Figs 5 and 6 to better expose the underlying mechanics of our method, and to showcase its value as a real-time inference tool for emerging epidemics with strong non-stationary transmission. These examples featured rapid changes, also known as events in information theory, caused by effective and ineffective countermeasures [3]. In all cases our metric responded rapidly whilst maintaining reliable, real-time incidence predictions. Such rapid, event-triggered responses (as in Fig 7) are known to be time-efficient [19]. A key objective of short-term forecasting is the speedy diagnosis of intervention efficacy. Our results confirmed the practical value of our method towards this objective. Importantly, we found in Figs 5 and 6 that the APE metric computed successively and causally during an unfolding outbreak, $k_s^*$, can quickly and correctly inform on changes in transmission and growth of an epidemic in real time.

While our metric is promising, we caution that much work remains to be done. Standard Poisson renewal models, such as those we have considered here, make several limiting assumptions including that (i) all cases are detected (i.e. there is no significant sampling bias), (ii) the serial interval and generation time distribution coincide and do not change over the epidemic lifetime and (iii) that heterogeneities in transmission within the infected population have negligible effect [10]. Our analysis of H1N1 influenza (1918) and SARS (2003) flagged some of these concerns. Comparing the $k^*$ to previously recommended weekly windows from [10], as in Figs 9 and 10, led to some interesting revelations.

Our APE approach selected a notably shorter window ($k^* = 2$ days) for both datasets. While this produced noisy $R_s$ estimates that seemed less reliable than those obtained with weekly windows, the predicted incidence values were central to understanding this discrepancy. The

weekly windows were considerably worse at forecasting the observed epi-curve, often systematically biased around the peak of the epidemic and sometimes predicting multimodal curves that were not reflected by the existing data. From a model selection perspective (among all EpiEstim-type models [10]), the shorter windows are therefore justified.

However, the noisy $R_t$ estimates are still undesirable. Both datasets are known to potentially contain super-spreading heterogeneities and other biases that may lead to outliers [10, 16]. Moving-average filters were applied in [16] to remove these artefacts from the H1N1 data. We used the same technique to regularise both datasets and then re-applied the APE. Results remained consistent, suggesting that if a Poisson model is valid then $k^* = 2$ is indeed the correct window choice. It may also be that the mean-variance equality of Poisson models is too restrictive for these datasets and so APE compensated for this inflexibility with short windows. A negative binomial renewal model where $I_t \sim \mathrm{NB}\left(\kappa, \frac{\Lambda_t R_t}{\Lambda_t R_t + \kappa}\right)$, with $\kappa$ as a noise parameter, would relax this equality yet still embody the Euler-Lotka dynamics of the original model. The extension of APE to these generalised models will form an upcoming study.

Real-time, model-based epidemic forecasts are quickly becoming an integral prognostic for resource allocation and strategic intervention planning [20]. As global infectious disease threats elevate, model-supported predictions, which can inform decision making as an outbreak progresses, have become imperative. In the ongoing COVID-19 pandemic, for example, modelling is being extensively used to inform policy. However, much is still unknown about the fundamentals of epidemic prediction and this uncertainty has inspired some reluctance in public health applications [20]. As a result, recent studies have called for careful evaluation of the predictive capacity of models, and demonstrated the importance of having standard metrics to compare models and qualify their uncertainties [21, 22].

Here we have presented one such metric for use in real time. Short-term forecasts aid immediate decision making and are more actionable than long-term ones (which are less reliable) [20, 21]. While we focussed on renewal models, the general APE metric (see Fig 1) can select among diverse model types and facilitate comparisons of their short-term predictive capacity and reliability. Its direct use of posterior predictive distributions within an honest, causal framework allows proper and problem-specific inclusion of uncertainty [14]. Common metrics such as the mean absolute error lack this adaptation [21]. Given its information-theoretic links and demonstrated performance we hope that our APE-based method can serve as a rigorous benchmark for model-based forecasts and help refine ongoing estimates of transmission as outbreaks unfold.

## Supporting information

**S1 Text. Further details and more general definitions and equations under sub-headings: Epidemic renewal models and Prospective model selection.**
(PDF)

## Author Contributions

**Conceptualization:** Kris V. Parag.

**Formal analysis:** Kris V. Parag.

**Investigation:** Kris V. Parag.

**Methodology:** Kris V. Parag.

**Project administration:** Kris V. Parag.

**Software:** Kris V. Parag.

**Supervision:** Christl A. Donnelly.

**Validation:** Kris V. Parag, Christl A. Donnelly.

**Writing – original draft:** Kris V. Parag.

**Writing – review & editing:** Kris V. Parag, Christl A. Donnelly.

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
