## [Decision Letter · Decision Letter 0]

1 Mar 2020

Dear Dr Parag,

Thank you very much for submitting your manuscript "Using information theory to optimise epidemic models for real-time prediction and estimation" for consideration at PLOS Computational Biology. As with all papers reviewed by the journal, your manuscript was reviewed by members of the editorial board and by several independent reviewers. The reviewers appreciated the attention to an important topic. Based on the reviews, we are likely to accept this manuscript for publication, providing that you modify the manuscript according to the review recommendations.

I congratulate the author's on a very nice manuscript that was well received by the reviewers. Both reviewers were complimentary of the methods, though they raised a number of concerns about the presentation. I would encourage the authors to revisit the figures, as both reviewers found that they were challenging to interpret. Specifically:

1. Please increase the font size for figure labeling

2. If possible, consider limiting the number of scenarios presented in the multi-panel figures (particularly Fig 2). This could be reduced to a smaller number of exemplary scenarios (e.g. a,f,c,e where the latter two reflect scenarios exemplary of interventions)

3. Figure 1 is quite dense and I agree with R2 that breaking this up into panels that could be labelled, and this referenced in the legend, would help with interpretation.

4. consider using a color palettes that have greater contrasts.

Both reviewers comment on the tendency for k* to be estimated at the minimal (both reviewers) or maximal (R2) values. Please elaborate on this phenomenon.

Please consider the suggestion 1 of R2. If it is possible to clarify the applicability in real-time, I suspect that would appeal to readers.

Please address the minor comments of R2 and comments 1&2 of R1.

Sincerely,

Matthew (Matt) Ferrari

Associate Editor

PLOS Computational Biology

Virginia Pitzer

Deputy Editor

PLOS Computational Biology

[LINK]

I congratulate the author's on a very nice manuscript that was well received by the reviewers. Both reviewers were complimentary of the methods, though they raised a number of concerns about the presentation. I would encourage the authors to revisit the figures, as both reviewers found that they were challenging to interpret. Specifically:

1. Please increase the font size for figure labeling

2. If possible, consider limiting the number of scenarios presented in the multi-panel figures (particularly Fig 2). This could be reduced to a smaller number of exemplary scenarios (e.g. a,f,c,e where the latter two reflect scenarios exemplary of interventions)

3. Figure 1 is quite dense and I agree with R2 that breaking this up into panels that could be labelled, and this referenced in the legend, would help with interpretation.

4. consider using a color palettes that have greater contrasts.

Both reviewers comment on the tendency for k* to be estimated at the minimal (both reviewers) or maximal (R2) values. Please elaborate on this phenomenon.

Please consider the suggestion 1 of R2. If it is possible to clarify the applicability in real-time, I suspect that would appeal to readers.

Please address the minor comments of R2 and comments 1&2 of R1.

Reviewer's Responses to Questions

**Comments to the Authors:**

Reviewer #1: This was an interesting paper that adds to the suite of methods for analysing case report data. In particular it provides a useful measure of the “window size” over which the reproductive ratio Rt should be calculated. As such, this could prove to be a powerful tool.

On the whole I found the paper to be extremely well written, and I only have a few minor comments.

1. In the abstract, I find the phrase “most succinctly describes” to be rather vague and confusing, could the authors seek a more informative description?

2. Page 9, the choice of a Gamma distribution for the posterior should be better motivated.

3. Figures: I found these hard to read. I realise that most people view on-line and therefore can greatly magnify figures, but a printed version is unreadable. In figure 7, I wonder if plotting Rt on a log-scale would help?

4. The paper ends on a whimper, with it being unclear if the k=2 found from real data is true or an artifact of a noisy system. I’m surprised that the authors didn’t test the method against simulations that were made sequentially noisier, this would seem an obvious test. I also feel that the speculation in the last paragraph of the results would sit far better in the discussion.

Reviewer #2: See attached

**Have all data underlying the figures and results presented in the manuscript been provided?**

Reviewer #1: Yes

Reviewer #2: Yes

PLOS authors have the option to publish the peer review history of their article (what does this mean?). If published, this will include your full peer review and any attached files.

Reviewer #1: No

Reviewer #2: No
---

## [Editor Report · Decision Letter 1]

13 May 2020

Dear Dr Parag,

Thank you very much for submitting your manuscript "Using information theory to optimise epidemic models for real-time prediction and estimation" for consideration at PLOS Computational Biology. As with all papers reviewed by the journal, your manuscript was reviewed by members of the editorial board and by several independent reviewers. The reviewers appreciated the attention to an important topic. Based on the reviews, we are likely to accept this manuscript for publication, providing that you modify the manuscript according to the review recommendations.

I thank the authors for their careful revisions. I do not see the need to send this back to review, but I would ask the authors to correct the legend for Figure 8 before I recommend this for publication. I assume the "(solid blue, left y axis)" and "(dotted red, right y axis)" refer to the points in the figure, but it isn't clear if I am assuming correctly or if there are elements missing from the figure. With this clarified, I will submit a final recommendation of "accept".

Sincerely,

Matthew (Matt) Ferrari

Associate Editor

PLOS Computational Biology

Virginia Pitzer

Deputy Editor

PLOS Computational Biology

[LINK]

I thank the authors for their careful revisions. I do not see the need to send this back to review, but I would ask the authors to correct the legend for Figure 8 before I recommend this for publication. I assume the "(solid blue, left y axis)" and "(dotted red, right y axis)" refer to the points in the figure, but it isn't clear if I am assuming correctly or if there are elements missing from the figure. With this clarified, I will submit a final recommendation of "accept".
---

## [Editor Report · Decision Letter 2]

27 May 2020

Dear Dr Parag,

We are pleased to inform you that your manuscript 'Using information theory to optimise epidemic models for real-time prediction and estimation' has been provisionally accepted for publication in PLOS Computational Biology.

Best regards,

Matthew (Matt) Ferrari

Associate Editor

PLOS Computational Biology

Virginia Pitzer

Deputy Editor

PLOS Computational Biology

I thank the authors for their attention the reviewer comments and commend them on a fine manuscript.

---

## [Editor Report · Acceptance letter]

24 Jun 2020

PCOMPBIOL-D-20-00012R2 

Using information theory to optimise epidemic models for real-time prediction and estimation

Dear Dr Parag,

I am pleased to inform you that your manuscript has been formally accepted for publication in PLOS Computational Biology. Your manuscript is now with our production department and you will be notified of the publication date in due course.

With kind regards,

Sarah Hammond
